# UNSEEN CLASS DISCOVERY IN OPEN-WORLD CLASSIFICATION

## ABSTRACT

This paper concerns open-world classification, where the classifier not only needs to classify test examples into seen classes that have appeared in training but also reject examples from unseen or novel classes that have not appeared in training. Specifically, this paper focuses on discovering the hidden unseen classes of the rejected examples. Clearly, without prior knowledge this is difficult. However, we do have the data from the seen training classes, which can tell us what kind of similarity/difference is expected for examples from the same class or from different classes. It is reasonable to assume that this knowledge can be transferred to the rejected examples and used to discover the hidden unseen classes in them. This paper aims to solve this problem. It first proposes a joint open classification model with a sub-model for classifying whether a pair of examples belongs to the same or different classes. This sub-model can serve as a distance function for clustering to discover the hidden classes of the rejected examples. Experimental results show that the proposed model is highly promising.

## 1 INTRODUCTION

An assumption made by classic supervised learning is that all classes appear in the test data must have appeared in training. This is called the *closed-world* assumption (Bendale & Boult, 2015; Scheirer et al., 2013). Although this assumption holds in many applications, it may be violated in dynamic and open environments. For example, in open-world object recognition, new objects may appear constantly and a classifier built from examples of old objects may incorrectly classify a new object as one of the old objects. This situation calls for *open-world classification* or simply *open classification*, which can classify those examples from the seen classes (appeared in training) and also detect/reject examples from unseen or novel classes (not appeared in training).

Ideally, an open classification system should be able to (1) assign each incoming/test example to one of the *seen classes* (appeared in training) and reject those examples from hidden *unseen classes* (not appeared in training), (2) discover hidden unseen classes in the rejected examples, and (3) learn the new classes incrementally. Since many existing methods exist for (1) and (3) (Scheirer et al., 2013; 2014; Bendale & Boult, 2015; Fei & Liu, 2016; Bendale & Boult, 2016), we will not study them in this paper. This paper focuses on (2). It will also employs a latest algorithm for (1) (Shu et al., 2017) as (2) is after (1) and we want to test our proposed technique for (2) based on the rejected examples from (1). To our knowledge, none of the existing open classification systems can do (2).

The key idea of the proposed technique for task (2), called *unseen class discovery*, is to transfer the class similarity knowledge learned from the seen classes to the hidden unseen classes. The transferred similarity knowledge is then used by a hierarchical clustering algorithm to cluster the rejected examples produced by an open classification model for (1) to discover the hidden classes in the rejected examples. Note that this transfer of knowledge is from supervised learning to unsupervised learning. It is different from traditional transfer learning as, in traditional transfer learning, knowledge is transferred from a supervised learning task to another supervised learning task or from an unsupervised learning task to another unsupervised learning task (Pan & Yang, 2010).

This proposed transfer is warranted because we human beings seem to group things based on our prior knowledge of what might be considered similar or different. For example, if we are given two objects and are asked whether they are of the same class/category or of different classes given some context, most probably we can tell. Why is that the case? We believe that we have learned in the

past what are considered to be of the same class or of different classes in a knowledge context. The knowledge context here is important. For example, we have learned to recognize some breeds of dogs, which forms the knowledge context. When we are given two new/unseen breeds of dogs, we probably know that they are of different breeds. If we are given many different dogs from each of the two breeds, we probably can cluster them into two clusters. However, if our previous knowledge only has classes such as dog, chicken, pig, cow, and sheep and we are given two different but unseen breeds of dogs, we probably will say that they are of the same kind/class and are dogs. However, if we are given a tiger and a rabbit, we will probably tell that they are from different classes.

**Problem Statement**: Given the labeled training data $D = \{(\mathbf{x}_1, y_1), (\mathbf{x}_i, y_2), \ldots, (\mathbf{x}_n, y_n)\}$ of $m$ seen classes, where $\mathbf{x}_i$ is the $i$-th example $y_i \in \{s_1, \ldots, s_m\} = \mathcal{S}$ is $\mathbf{x}_i$'s class label, we want to (a) build a model $f(\mathbf{x})$ that can classify each test example $\mathbf{x}$ to one of the $m$ seen classes in $\mathcal{S}$ or reject it as unseen, (b) build a binary classification model $g(\mathbf{x}_p, \mathbf{x}_q)$ which can tell if any two test examples ($\mathbf{x}_p$ and $\mathbf{x}_q$ from seen or unseen classes) belong to the same class or not, and (c) for all rejected test examples, discover how many hidden classes they belong to based on $g(\mathbf{x}_p, \mathbf{x}_q)$.

To solve this problem, we first propose a model that is a combination of two main neural networks: an Open Classification Network (OCN) for seen class classification and unseen class rejection, and a Pairwise Classification Network (PCN) that learns a binary classifier to predict whether two given examples come from the same class or different classes, i.e., $g(\mathbf{x}_p, \mathbf{x}_q)$. The two networks share the same representation learning component. The training data for OCN is the original seen class training data. The positive training data of PCN consists of a set of pairs of intra-class (same class) examples, and the negative training data consists of a set of pairs of inter-class (different classes) examples all from seen classes. A hierarchical clustering method then uses the function $g(\mathbf{x}_p, \mathbf{x}_q)$ (which can be regarded as a distance function) to find the number of hidden classes (clusters) in the unseen/rejected class examples. Experimental results show that the proposed technique is highly promising.

## 2 RELATED WORK

Several prior works exist on open classification that can reject unseen class examples (Scheirer et al., 2013; Jain et al., 2014; Fei & Liu, 2016; Scheirer et al., 2014; Jain et al., 2014; Dalvi et al., 2013). Recently, two deep learning approaches were also proposed. One is called *OpenMax* (Bendale & Boult, 2016) and the other DOC (Shu et al., 2017). Our OCN module uses DOC, which was for open text classification, but our experiments show that it also works well for images and outperforms OpenMax. After classification, (Bendale & Boult, 2015) and (Fei et al., 2016) also incrementally learn the unseen classes in the rejected examples after manually labeling. De Rosa et al. (2016) and Rebuffi et al. (2016) proposed some algorithms only for incremental learning. Our work does not focus on incremental learning. Fu & Sigal (2016) solves a similar problem of detecting unseen class examples and classifying them by leveraging image description vocabulary and assuming unseen class examples' labels are available in the vocabulary. We do not use any description vocabulary.

The proposed pairwise classification network can be regarded as learning a distance function. It is thus related to metric learning (Xing et al., 2002; Yang & Jin, 2006; Bellet et al., 2013; Martin & Jurafsky, 2000; Manning et al., 2008; Severyn & Moschitti, 2015; Guo et al., 2016). However, the existing works learn distance functions for only seen class clustering or information retrieval, but we learn for unseen classes clustering. None of these works find the number of clusters. Hsu et al. (2016) proposed a deep clustering network that can be transferred to a different domain. However, their work cannot find the number of clusters, which is the key task of our work.

Our problem is related to non-exhaustive learning (Dundar et al., 2012; Akova et al., 2012). Two Bayesian non-parametric models were proposed to identify mixture components or clusters in the test data that may cover unseen classes. However, their clusters are not classes. In fact, multiple clusters map to one class, and the mapping is done manually. This is clearly not suitable for our work. Our setting requires that one cluster is one class. We can detect precise number of unseen classes.

Our work is also different from semi-supervised clustering (Yi et al., 2013; 2015). Semi-supervised clustering uses must-links and cannot-links or pairwise constraints of data points provided by the user to improve clustering accuracy on the same data. Our PCN model is learned from seen class pairs as a distance or similarity metric, which is applied to rejected example pairs (unseen class pairs) as a clustering metric. Semi-supervised clustering also does not find the number of clusters.

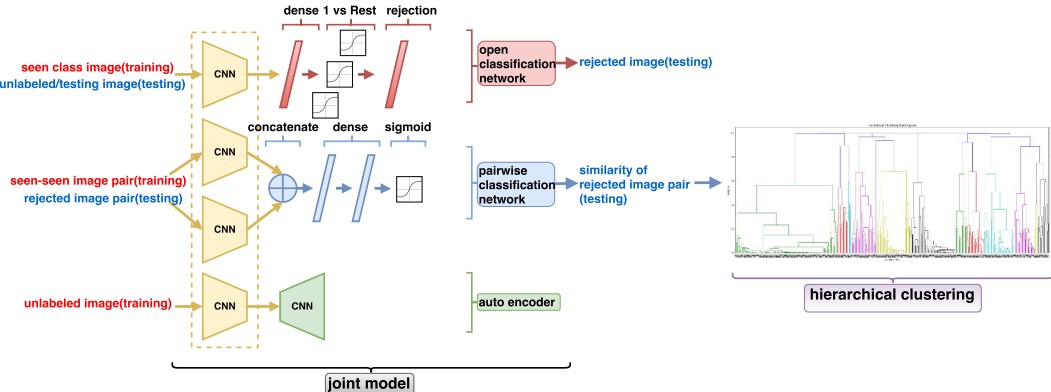

Figure 1: Overall framework: Open Classification Network (OCN), Pairwise Classification Network (PCN) and auto-encoder are on the left and a hierarchical clustering process is on the right. The left 3 components are jointly trained. Data in red color and blue color are training data and testing data respectively. Rejected images are testing data that are rejected by OCN as unseen class examples and then rejected image pairs are fed into PCN to compute distances for hierarchical clustering.

## 3 THE PROPOSED SYSTEM

The proposed system has four (4) components: an Open Classification Network (OCN), a Pairwise Classification Network (PCN), an auto-encoder, and a hierarchical clustering method (see Fig. 1).

(1) OCN is used for open classification (traditional classification with rejection capability), which can produce rejected examples when tested on both seen and unseen class examples.

(2) PCN classifies whether two input examples are from the same class or different classes.

(3) Auto-encoder is used to learn representations from unlabeled examples.

(4) Hierarchical clustering clusters the rejected examples from OCN using PCN as the distance function. It gives the number of hidden clusters or classes embedded in the rejected examples.

The representation learning part of the first 3 components are shared. We use a multi-layer CNN (Masci et al., 2011). The hyper-parameters of CNN are discussed in the Experiment section. We jointly train the 3 components together as a multi-task learning model (Pan & Yang, 2010). The joint loss is the sum of 3 components' losses. The whole system works as follows:

**(1) Training Phase**: we form three kinds of training datasets: (i) open-classification dataset on seen classes for OCN, (ii) pairwise classification dataset from seen classes for PCN, and (iii) all unlabeled class examples for auto-encoder. We jointly train 3 components using the above datasets.

**(2) Testing/Predicting Phase**: we let OCN predict on the test dataset including unlabeled examples from both seen and unseen classes. Collect the rejected examples by OCN for the clustering phase.

**(3) Clustering Phase**: we form pairwise examples from the rejected examples and feed them into PCN. The prediction results of PCN are used as distances in hierarchical clustering to cluster rejected examples into clusters. The important point about this clustering process is that it can automatically find the number of clusters or classes.

### 3.1 OPEN CLASSIFICATION NETWORK

As noted earlier, our focus is not on open classification, but on identifying the hidden classes of the rejected examples. However, in order to test our hidden class discovery algorithm, we need a system to produce rejected examples. In our case, we use the latest DOC algorithm (Shu et al., 2017) for our OCN network. Although it was designed for open text classification, it also performs well on images and is significantly better than the latest OpenMax method for open image classification (Bendale & Boult, 2016). OCN contains a CNN representation learning part shared with the other networks, followed by a fully-connected layer and a 1-vs-rest layer of sigmoid functions. It does not use the

usual softmax as the output layer (Goodfellow et al., 2016; Bendale & Boult, 2016) as the softmax function does not have the rejection capability given its normalized probability distribution on seen classes, and thus is more suitable for closed-world classification.

The 1-vs-rest layer naturally can reject unseen class examples because if a test example cannot be classified to any existing/seen class based on the 0.5 probability threshold, it should be rejected. However, DOC does not use 0.5 as the threshold for each seen class. It uses a much bigger probability threshold computed based on the idea of outlier detection in the context of Gaussian distribution. Interested readers can refer to the original paper for details.

## 3.2 PAIRWISE CLASSIFICATION NETWORK

Next, we introduce Pairwise Classification Network (PCN) to learn intra-class and inter-class difference from the seen classes using a binary classification model, which is later transferred to unseen classes as a guidance for uncovering unseen classes using clustering. PCN has two identical branches of CNNs, which are concatenated and followed by two fully-connected layers and a simple sigmoid function to predict if two examples from two branches are from the same class. We feed pairs of seen class examples into two branches to train PCN.

As discussed earlier, the positive training data consists of a set of pairs of intra-class (same class) examples, and the negative training data consists of a set of pairs of inter-class (different classes) examples all from seen classes. When the number of labeled examples $n$ from the seen classes is large, it is infeasible to exhaust all pairs of intra-class and inter-class examples since the number of pairs will grow at $O(n^2)$. This dramatically increases the time spent on training. Instead, we sample pairs of examples from seen classes and keep the numbers of pairs for both the intra-class examples and the inter-class examples the same. Each pair of intra-class examples or inter-class examples are uniformly sampled from $\mathcal{S}$ (which is the set of seen classes). We discuss the details of sampling in the Experiment section. We use $\mathcal{L}_{PCN}$ to denote the binary log loss over all sampled pairs.

## 3.3 JOINT TRAINING

We further introduce an auto-encoder to simultaneously learn unsupervised representations for examples from unlabeled examples, because purely using examples from the seen classes can easily overfit the seen classes and lead to bad representations for unseen classes. We use $\mathcal{L}_{ae}$ to denote the loss for auto-encoder. We jointly train OCN, PCN and the auto-encoder as a multi-task learning process and the total loss of this joint optimization is:

$$\mathcal{L}_{joint} = \mathcal{L}_{OCN} + \mathcal{L}_{PCN} + \mathcal{L}_{ae},$$

$$\mathcal{L}_{OCN} = \sum_{i=1}^{n} \sum_{j=1}^{m} -\mathbb{I}\{y_i = s_j\} \log p(\hat{y}_i = s_j) - \mathbb{I}\{y_i \neq s_j\} \log(1 - p(\hat{y}_i = s_j)),$$

$$\mathcal{L}_{PCN} = \sum_{i=1}^{n} -\mathbb{I}\{y_i = 1\} \log p(\hat{y}_i = 1) - \mathbb{I}\{y_i = 0\} \log p(\hat{y}_i = 0), \mathcal{L}_{ae} = \sum_{i=1}^{n} \| \mathbf{x}_i - \hat{\mathbf{x}}_i \| .$$

$$(1)$$

## 3.4 HIERARCHICAL CLUSTERING AND VALIDATION CLASSES

We use hierarchical clustering (Gowda & Krishna, 1978) to find the number of clusters in the rejected examples and identify clusters. The core idea in hierarchical clustering is to merge two clusters at each step until some stopping condition is met based on a distance function $d(\cdot, \cdot)$. The output of PCN can naturally be used as such a distance function as it is trained to separate the same and different class instance/example pairs. We use complete-linkage hierarchical clustering. Its distance between 2 clusters $C_1$ and $C_2$ is the maximum distance between the examples in the two clusters:

$$d(C_1, C_2) = \max_{\mathbf{x}_p \in C_1, \mathbf{x}_q \in C_2} g(\mathbf{x}_p, \mathbf{x}_q). \tag{2}$$

The advantage of hierarchical clustering is that it does not require a pre-defined number of clusters, which is suitable for our problem as the number of clusters is unknown in the open environment.

However, since we need the number of clusters (or hidden unseen classes), a distance threshold $\theta$ is required as a stopping criteria to stop further merging of clusters. The sigmoid function in PCN naturally has an assumed threshold $\theta = 0.5$ as the stopping criterion. However, this threshold may only be suitable for seen classes but not unsuitable for unseen classes. We thus hold out some classes from the seen classes as the validation classes $V$ to determine a better $\theta$ for hierarchical clustering. When hierarchical clustering reaches the true number of clusters $|V|$, we use the largest distance between every two clusters $C_i$ and $C_j$ as the threshold $\theta$:

$$\theta = \max_{i<=|V|, j<=|V|, i\neq j} d(C_i, C_j).$$ (3)

Clustering of the rejected examples stops if the pair of clusters to be merged next has the distance greater than or equal to $\theta$. Note that $\theta$ is not easily used in partitioning clustering methods such as $K$-means because they require the number of clusters as input.

**An alternative Approach:** In the above, we use PCN as a distance function and find a threshold $\theta$ for clustering. But following the same approach, we can also directly use cosine or Euclidean distance to perform hierarchical clustering and find a threshold for determining the number of clusters by clustering the training data of seen classes. We experimented with this approach, but it worked poorly. The reason is that the pairwise similarities for different classes are all over the map and it is very hard to find a cutoff threshold using the seen class data that is applicable to the unseen class data. The effect of this problem is significantly reduced by using PCN because PCN serves as a nonlinear scaling function that standardizes the pairwise similarities of examples in different classes.

## 4 EXPERIMENTS

We perform evaluation of the proposed technique for discovering the number of novel/unseen classes using two publicly available datasets: MNIST and EMNIST

(1) **MNIST**[1]: MNIST is a well-known database of handwritten digits (10 classes), which has a training set of 60,000 examples, and a test set of 10,000 examples. We use 6 classes as the set of seen classes and use the rest 4 classes as unseen classes (all randomly chosen). We use the same validation classes from the following EMNIST dataset as the validation dataset for MNIST.

(2) **EMNIST**[2](Cohen et al., 2017): EMNIST is an extension of MNIST to commonly used characters such as English alphabet. It is derived from the NIST Special Database 19. We use EMNIST Balanced dataset with 47 balanced classes. It has a training set of 112,800 examples and a test set of 18,800 examples. We use 33 classes as the set of seen classes, 10 classes as the unseen classes and 4 classes as the validation seen classes (again, all randomly chosen).

### 4.1 HYPER-PARAMETER SETTINGS

For CNN, we use two convolution layers (with 128 filters of size 3x3 and 64 filters of size 3x3, resp.) with one max-pooling layer (of size 2x2), followed by another convolution layer (with 32 filters of size 3x3) and a max-pooling layer (of size 2x2). The fully-connected layer before 1-vs-rest layer is of size 1024. For PCN, the fully-connected layer before the pairwise classification layer is of size 256. For the decoder of auto-encoder, we use one convolution layer (with 32 filters of size 3x3), one up-sampling layer (of size 2x2), another convolution layer (with 64 filters of 3x3) and up-sampling layer (of size 2x2), followed by two convolution layers (with 128 filters of size 3x3 and 1 filter of size 3x3, resp.). We leverage Adam optimizer (Kingma & Ba, 2014) to optimize the loss function Eq. 1 and empirically set the learning rate as 0.005, $\beta_1$ as 0.9, $\beta_2$ as 0.999 and $\epsilon$ as $e^{-8}$. The dropout rate is set as 0.25 to avoid overfitting. The mini-batch size is set as 256.

### 4.2 OPEN CLASSIFICATION NETWORK (OCN)

We first describe the experiments on OCN, based on DOC and originally designed for open text classification (Shu et al., 2017) and Openmax, originally designed for open image classification (Bendale

---

[1] http://yann.lecun.com/exdb/mnist/
[2] https://www.nist.gov/itl/iad/image-group/emnist-dataset

| algorithm | MNIST | | | | EMNIST | | | |
|---|---|---|---|---|---|---|---|---|
| | $(m+1)$ classes | rejection class | | | $(m+1)$ classes | rejection class | | |
| | macro-$\mathcal{F}$ | $\mathcal{P}$ | $\mathcal{R}$ | $\mathcal{F}$ | macro-$\mathcal{F}$ | $\mathcal{P}$ | $\mathcal{R}$ | $\mathcal{F}$ |
| OCN | 0.914 | 0.920 | 0.824 | 0.869 | 0.832 | 0.664 | 0.47 | 0.554 |
| OpenMax(weibull=20) | 0.678 | 0.955 | 0.026 | 0.051 | 0.789 | 0.786 | 0.07 | 0.13 |
| OpenMax(weibull=1000) | 0.684 | 0.956 | 0.043 | 0.083 | 0.803 | 0.725 | 0.239 | 0.359 |

Table 1: Macro-$\mathcal{F}$ is average $F$-score on $m+1$ classes, where $m$ is the number of seen classes and 1 is the rejection class. $\mathcal{P}$, $\mathcal{R}$ and $\mathcal{F}$ are precision, recall and F-score of the rejection class only

| Type of Pair | MNIST | EMNIST |
|---|---|---|
| seen-seen | 0.994 | 0.965 |
| seen-unseen | 0.752 | 0.874 |
| unseen-unseen | 0.700 | 0.810 |

Table 2: Accuracy of pairwise classification (whether two examples are from the same class or not)

| algorithm | GT | Encoder + HC | | K-means NMI score | | PCN + HC | |
|---|---|---|---|---|---|---|---|
| | # of C | # of C | NMI | K from GT | K from PCN+HC | # of C | NMI |
| MNIST | 4 | 3 | 0.414 | 0.710 | 0.66 | 5 | 0.302 |
| EMNIST | 10 | 4 | 0.479 | 0.683 | 0.683 | 10 | 0.583 |

Table 3: Clustering results of unseen classes on MNIST and EMNIST. GT means the ground truth number of clusters for unseen classes, and # of C means the number of clusters.

| algorithm | GT | OCN+Encoder+HC | | K-means NMI score | | OCN+PCN+HC | |
|---|---|---|---|---|---|---|---|
| | # of C | # of C | NMI | K from GT | K from OCN+PCN+HC | # of C | NMI |
| MNIST | 4 | 4 | 0.478 | 0.563 | 0.591 | 6 | 0.320 |
| EMNIST | 10 | 4 | 0.312 | 0.586 | 0.543 | 14 | 0.500 |

Table 4: Clustering results of rejected examples on MNIST and EMNIST. GT means the ground truth number of clusters for unseen classes, and # of C means the number of clusters.

& Boult, 2016)[3]. Both systems are based on deep learning. OpenMax has a parameter (Weibull tail size, with 20 as default) that needs to be tuned using a validation set. We did that by following the original paper and using our validation dataset to search in the range from 0 to 2000, each step is 20. We found that Weibull tail size of 1000 gives the best results. DOC also has a parameter, but we just used its default value, which already performs quite well and markedly better than OpenMax

Note that for open classification, we treat unseen classes as one rejection class. We evaluate OCN using macro-$\mathcal{F}$ score of all classes, and precision $\mathcal{P}$, recall $\mathcal{R}$ and $F$-score $\mathcal{F}$ on the rejection class. The experimental results of OCN and OpenMax are shown in Table 1. For OpenMax, we report its scores under its default parameter value (Weibull tail size of 20) and the best parameter value (Weibull tail size of 1000). From the table, we observe that OCN has superior performance to OpenMax on rejection for both MNIST and EMNIST datasets. OCN overall has better precision than recall. The recall rates of OCN are significantly better than OpenMax, indicating many unseen examples are successfully rejected without mixing them into seen classes.

### 4.3 PAIRWISE DATA SAMPLING

We generate the training data for training the pairwise signal from seen classes as follows: For intra-class examples (each consisting of a pair of images belonging to the same class), we sample 10000/4000 (MNIST/EMNIST) pairs of images from each seen class and ensure that no duplicated pairs exist and no duplicated images exist in a pair. All these examples form the positive class data. For inter-class examples (each consisting of a pair of images from different classes), we uniformly sample 10000/4000 (MNIST/EMNIST) pairs of images and ensure that the first image and the second image are from different classes and also no duplicates exist. All these examples form the negative class data. Thus, the total number of training pairs (examples) is $m * (20000/8000)$

---

[3]https://github.com/abhijitbendale/OSDN

(MNIST/EMNIST), where $m$ is the number of seen classes. These pairs are called training *seen-seen pairs* to distinguish them from testing pairs.

For testing data, besides sampling testing seen-seen pairs in the same way as training seen-seen pairs, we also sample testing seen-unseen and testing unseen-unseen pairs. (1) Testing seen-seen pairs: these pairs are sampled in the same way to evaluate the pairwise classification performance on seen classes. (2) Testing seen-unseen pairs: we sample 1000 pairs such that the first image is from a seen class and the second image from an unseen class. The total number of seen-unseen pairs is $m * 1000$ and they are all inter-class pairs. (3) Testing unseen-unseen pairs: these pairs are similar to seen-seen pairs but are generated from unseen classes. The number of unseen-unseen pairs is $l * 2000$, where $l$ is the number of unseen classes.

## 4.4 Pairwise Classification Network

We evaluate PCN on testing seen-seen pairs, seen-unseen pairs, and unseen-unseen pairs to test its capability of pairwise (binary) classification on seen and unseen classes and use accuracy as the evaluation metric. Table 2 shows the experimental results. We can see that overall PCN has a good performance on transferring from seen classes to unseen classes. Overall PCN on EMNIST has better performance than MNIST because EMNIST has more seen classes to obtain better prior knowledge on classes. The accuracy on seen-unseen pairs is better than on unseen-unseen pairs since PCN has better knowledge of seen classes obtained from training. The accuracy on unseen-unseen pairs is also relatively good even though PCN has no supervised signal on unseen classes, which means that we can use such pairwise signals to guide the clustering procedure discussed below.

## 4.5 Clustering

Based on the success of pairwise classification on unseen classes, we pipeline PCN with hierarchical clustering (HC) (Gowda & Krishna, 1978) together to cluster the rejected images. Here the predictions of pairwise classification are used as the distance function $d(\cdot, \cdot)$ for clustering. To find an suitable threshold $\theta$ (stopping criterion) for determining the number of clusters, we use the 4 validation classes from EMNIST to run PCN and HC to generate the best cutoff distance threshold $\theta$, which is then used as the stopping criterion in clustering to determine the number of clusters or unseen classes in the rejected examples. We experiment with two clustering settings.

**(1) Clustering Unseen Classes:** We assume that open classification can reject unseen examples perfectly (100%). That is, we use the ground truth of unseen class examples to test the clustering performance. We refer to this experiment as PCN+HC (as OCN is not used).

**(2) Clustering the Rejected Examples:** This is the realistic situation and the pipelined performance of OCN+PCN+HC. The examples to be clustered are all the rejected examples from OCN, which contain wrongly rejected seen class examples.

**Evaluation Measures:** To fully test our method, we perform two kinds of evaluation:

(1) Number of clusters: We compare the number of discovered clusters and the true number of clusters in the unseen class test data.

(2) Quality of clusters: Here we compare the cluster membership of the test images with these images' ground-truth labels using the popular evaluation metric (we re-defined some notations here): *Normalized Mutual Information* (NMI) (Pluim et al., 2000), which is a normalization of the Mutual Information (MI) to scale the results to between 0 (no mutual information) and 1 (perfect correlation). Given cluster $\mathcal{C}$ and ground truth label $\mathcal{Y}$:

$$\textbf{NMI}(\mathcal{C}, \mathcal{Y}) = \frac{I(\mathcal{C}, \mathcal{Y})}{\sqrt{H(\mathcal{C})H(\mathcal{Y})}}, I(\mathcal{C}, \mathcal{Y}) = \sum_i \sum_j \frac{|c_i \cap y_j|}{N} \log\left(\frac{N|c_i \cap y_j|}{|c_i||y_j|}\right), \text{ and } H(\mathcal{C}) = -\sum_i \frac{|c_i|}{N} \log\left(\frac{|c_i|}{N}\right), \quad (4)$$

where $N$ is the total number of images, $H(\cdot)$ is the entropy and $I(\cdot, \cdot)$ is the mutual information.

### 4.5.1 Baselines:

For clustering of unseen class or rejected examples, we compare our method with two baselines **Encoder+HC** and **K-means**. We compare with Encoder+HC in terms of both measures above. But for K-means, which requires a pre-specified number of clusters $K$, we can only compare the cluster

quality of K-means given that the number of clusters is known. Although our main task is to find the number of clusters in the rejected examples, if we want to incrementally learn the new/unseen classes we need to cluster the rejected examples well (each cluster is a class), for which K-means has an advantage over our hierarchical clustering as we will see shortly.

**Encoder+HC:** We use the shared CNN encoder part (without the component of pairwise classification) together with hierarchical clustering in this baseline to demonstrate that pairwise classification is effective. We use the Euclidean distance of two images' representations as the clustering metric. This baseline also takes advantage of validation classes to decide a threshold $\theta$ for stopping.

**K-means:** We use the classic K-means (MacQueen et al., 1967) to cluster the representations of examples from the shared CNN encoder part. K-means requires a pre-defined number of clusters $K$. We use the following 4 numbers of clusters to evaluate K-means: (1) the ground truth number of unseen classes ($K$ from GT in Table 3); (2) the estimated number of clusters from PCN+HC on Clustering Unseen Classes experiment ($K$ from PCN + HC in Table 3); (3) the number of true clusters/classes in the rejected examples that may contain both seen and unseen classes ($K$ from GT in Table 4); (4) the estimated number of clusters in the rejected examples from OCN+PCN+HC on Clustering Rejected Examples experiment ($K$ from OCN+PCN+HC in Table 4).

### 4.5.2 Experiment Results and Analysis:

The experimental results on the Clustering Unseen Classes experiment is shown in Table 3, where MNIST and EMNIST have 4 and 10 unseen classes, respectively. We can see that PCN+HC can almost find the correct number of clusters. But without PCN, the baseline Encoder+HC cannot find the correct number of clusters on EMNIST, which shows the importance of PCN.

The results on the Clustering Rejected Examples experiment are shown in Table 4. In this case, the rejected examples are noisy (including some seen class examples). However, the estimated number of clusters is very close to the ground truth number of unseen classes. This means that the number of clusters predicted by OCN+PCN+HC is not much affected by incorrectly rejected seen examples, which are the minority classes in the rejected examples. OCN+PCN+HC has a better performance on EMNIST than on MNIST because OCN+PCN+HC can obtain more prior knowledge about classes from EMNIST's larger number of seen classes.

The reason that HC (hierarchical clustering) using PCN can find good number of clusters is that PCN serves as a nonlinear scaling function that standardizes different class similarities and thus the threshold obtained from the validation set can be effectively transferred to the unseen classes. However, it is poorer than $K$-means in terms of NMI given the number of discovered clusters (to $K$-means) because the PCN scaling for classification also made the distances between two examples somewhat distorted. However, $K$-means cannot find the number of clusters, which is our key task. Thus, in practice we can use hierarchical clustering and PCN to find the number of clusters from the rejected examples and then use $K$-means to form the best clusters using the number of discovered clusters as the input $K$.

## 5 Conclusion

As learning is increasingly used in dynamic and open environments, it should no longer make the closed-world assumption. Open-world learning is required which can not only classify examples from the seen classes but also reject examples from unseen classes. What is also important is to identify the hidden unseen classes from the reject examples, which will enable the system to learn these new classes automatically rather than requiring manual labeling. This paper first proposed a joint model for performing open image classification and for predicting whether two images are from the same class using only the seen class training data. This latter capability enables the transfer of class distance/similarity knowledge from seen classes to unseen classes, which is exploited by a hierarchical clustering algorithm for discovering the number of hidden classes in the rejected examples. Our experiments demonstrated the effectiveness of the proposed approach.

ACKNOWLEDGMENTS

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
