# OpenReview forum: "Unseen Class Discovery in Open-world Classification"
_ICLR.cc/2018/Conference — Reject_

### Official Review · AnonReviewer1 · 2017-11-23
**Interesting idea, yet should be explored more extensively**

**Rating:** 5
**Confidence:** 4

**Review:**

The main goal of this paper is to cluster images from classes unseen during training.
This is an interesting extension of the open-world paradigm, where at test time, the classifier has to identify images beloning to the C seen classes during training, but also identify (reject) images which were previously unseen. These rejected images could be clustered to identify the number of unseen classes; either for revealing the underlying structure of the unseen classes, or to reduce annotation costs.

In order to do so, an extensive framework is proposed, consisting of 3 ConvNet architectures, followed by a hierarchical clustering approach. The 3 ConvNets all have a different goal:
1. an Open Classification Network (per class sigmoid, trained 1vsRest, with thresholds for rejection)
2. Pairwise Classification Network, (binary sigmoid, trained on pairs of images of same/different classes)
3. Auto encoder network

These network are jointly trained, and the joint-loss is simply the addition of a cross-entropy loss (from OCN), the binary cross-entropy loss (from PCN) and a pixel wise loss (from AE).
Remarks:
- it is unclear if the ConvNet weights of the first layers are shared).
- it is unclear how joint training might help, given that the objectives do not influence each other
- Eq 1:
  *label "y_i" has two different semantics (L_ocn it is the class label, while in L_pcn it is the label of an image pair being from the same class or not)
  * s_j is undefined
  * relation between the p(y_i = 1) (in PCN) and g(x_p,x_q) in Eq 2 could be made more explicit, PCN depends on two images, according to Eq 1, it seems just a sum over single images.
- It is unclear why the Auto Encoder network is added, and what its function is.
- It is unclear wether OCN requires/uses unseen class examples during training.
- Last paragraph of 3.1 "The 1-vs-rest ... rejected", I don't see why you need 1vsRest classifiers for this, a multi-class (softmax) output can also be thresholded to reject an test image from the known classes and to assign it to the unknown class.


Experimental evaluation
The experimental evaluation uses 2 datasets, MNIST and EMNIST, both are very specific for character recognition. It is a pity that not also more general image classification has been considered (CIFAR100, ImageNet, Places365, etc), that would provide insights to the more general behaviour of the proposed ideas.

My major concern is that the clustering task is not extensively explored. Just a single setting (with a single random sampling of seen/unseen classes) has been evaluated. This is -in part- due to the nature of the chosen datasets, in a 10 class dataset it is difficult to show the influence of the number of unseen classes. So, I'd really urge the authors to extend this evaluation. Will the method discover more classes when 100 unknown classes are used? What kind of clusters are discovered? Are the types of classes in the seen/unseen classes important, I'd expect at least multiple runs of the current experiments on (E)MNIST.

Further, I miss some baselines and ablation study. Questions which I'd like to seen answered: how good is the OCN representation when used for clustering compared to the PCN representation? What is the benefit of joint-training? How important is the AE in the loss?

Remaining remarks
- Just a very simple / non-standard ConvNet architecture is trained. Will a ResNet(32) show similar performance?
- In Eq 4, |C_i || y_j| seems a strange notation for union.

Conclusion
This paper brings in an interesting idea, is it possible to cluster the unseen classes in an open-world classification scenario?  A solution using a pairwise convnet followed by hierarchical clustering is proposed. This is a plausible solution, yet in total I miss an exploration of the solution.

Both in terms of general visual classification (only MNIST is used, while it would be nice to see results on CIFAR and/or ImageNet as in Bendale&Boult 2016), as in exploration of different scenarios (different number of unseen classes, different samplings) and ablation of the method (independent training, using OCN for hierarchical clustering, influence of Auto Encoder). Therefore, I rate this paper as a (weak) reject: it is just not (yet) good enough for acceptance.

---

### Official Review · AnonReviewer3 · 2017-11-25
**targeting a synthesized problem**

**Rating:** 5
**Confidence:** 5

**Review:**

This paper concerns open-world classification.  The open-world related tasks have been defined in many previous works. This paper had made a good survey.
The only special point of the open-word classification task defined in this paper is to employ the constraints from the similarity/difference expected for examples from the same class or from different classes.  Unfortunately, this paper is lack of novelty.

Firstly, the problem context and setting is kinda synthesized. I cannot quite imagine in what kind of applications we can get “a set of pairs of intra-class (same class) examples, and the negative training data consists of a set of pairs of inter-class”.

Secondly, this model is just a direct combination of the recent powerful algorithms such as DOC and other simple traditional models. I do not really see enough novelty here.

Thirdly, the experiments are only on the MNIST and EMNIST; still not quite sure any real-world problems/datasets can be used to validate this approach.
I also cannot see the promising performance. The clustering results of rejected
examples are still far from the ground truth, and comparing the result with
a total unsupervised K-means is a kind of unreasonable.

---

### Official Review · AnonReviewer2 · 2017-11-27
**Interesting idea on transferring the similarity function learned from known classes but it basically seems like a clustering paper with limited empirical evidence of its value.**

**Rating:** 4
**Confidence:** 4

**Review:**

This paper focuses on the sub-problem of discovering previously unseen classes for open-world classification.
It employs a previously proposed system, Open Classification Network, for classifying instances into known classes or rejecting as belonging to an unseen class, and applies hierarchical clustering to the rejected instances to identify unseen classes.
The key novel idea is to learn a pairwise similarity function using the examples from the known classes to apply to examples of unknown classes. The argument is that we tend to use the same notion of similarity and dissimilarity to define classes (known or unknown) and one can thus expect the similarity function learned from known classes to carry over to the unknown classes.  This concept is not new. Similar idea has been explored in early 2000 by Finley and Joachims in their ICML paper titled "Supervised Clustering with Support Vector Machines".  But to the best of my knowledge, this is the first paper that applies this concept to the open world classification task.

Once we learn the similarity function, the rest of the approach is straightforward, without any particular technical ingenuity.  It simply applies hierarchical clustering on the learned similarities and use cross-validation to pick a stopping condition for deciding the number of clusters.
I find the experiments to be limited, only on two hand-written digits/letters datasets.  Such datasets are too simplistic. For example, simply applying kmeans to PCA features of the images on the MNIST data can get you pretty good performance.
Experiments on more complex data is desired, for example on Imagenet classes.

Also the results do not clearly demonstrate the advantage of the proposed method, in particular the benefit of using PCN. The number of clusters found by the algorithm is not particularly accurate and the NMI values obtained by the proposed approach does not show any clear advantage over baseline methods that do not use PCN.

Some minor comments:
When applied to the rejected examples, wouldn't the ground truth # of clusters no longer be 4 or 10 because there are some known-class examples mixed in?
For the base line Encoder+HC, was the encoder trained independently? Or it's trained jointly with PCN and OCN?  It is interesting to see the impact of incorporating PCN into the training of OCN and encoder. Does that have any impact on accuracy of OCN?
It seems that one of the claimed benefit is that the proposed method is effective at identifying the k. If so, it would be necessary to compared the proposed method to some classic methods for identifying k with kmeans, such as the elbow method, BIC, G-means etc, especially since kmeans seem to give much better NMI values.

---

### Decision · Program_Chairs · 2018-01-29
**ICLR 2018 Conference Acceptance Decision**

**Decision:**

Reject

**Comment:**

Three reviewers recommended rejection and there was no rebuttal to overturn their recommendation.